# Antifungal Resistance in *Candida auris*: Molecular Determinants

**DOI:** 10.3390/antibiotics9090568

**Published:** 2020-09-02

**Authors:** María Guadalupe Frías-De-León, Rigoberto Hernández-Castro, Tania Vite-Garín, Roberto Arenas, Alexandro Bonifaz, Laura Castañón-Olivares, Gustavo Acosta-Altamirano, Erick Martínez-Herrera

**Affiliations:** 1Unidad de Investigación, Hospital Regional de Alta Especialidad de Ixtapaluca, Estado de México, 56530 Mexico City, Mexico; magpefrias@gmail.com (M.G.F.-D.-L.); mq9903@live.com.mx (G.A.-A.); 2Departamento de Ecología de Agentes Patógenos, Hospital General “Dr. Manuel Gea González”, Ciudad de México, 14080 Mexico City, Mexico; rigo37@gmail.com; 3Unidad de Micología, Facultad de Medicina, Universidad Nacional Autónoma de México, 04510 Mexico City, Mexico; tania.vite.garin@gmail.com (T.V.-G.); lrcastao@unam.mx (L.C.-O.); 4Sección de Micología, Hospital General “Dr. Manuel Gea González”, Ciudad de México, 14080 Mexico City, Mexico; rarenas98@hotmail.com; 5Departamento de Micología, Servicio de Dermatología, Hospital General “Dr. Eduardo Liceaga”, Ciudad de México, 06720 Mexico City, Mexico; a_bonifaz@yahoo.com.mx

**Keywords:** antifungal resistance, *Candida auris*, amphotericin B, 5-fluorocytosine, caspofungin, fluconazole

## Abstract

Since *Candida auris* integrates strains resistant to multiple antifungals, research has been conducted focused on knowing which molecular mechanisms are involved. This review aims to summarize the results obtained in some of these studies. A search was carried out by consulting websites and online databases. The analysis indicates that most *C. auris* strains show higher resistance to fluconazole, followed by amphotericin B, and less resistance to 5-fluorocytosine and caspofungin. In *C. auris,* antifungal resistance to amphotericin B has been linked to an overexpression of several mutated *ERG* genes that lead to reduced ergosterol levels; fluconazole resistance is mostly explained by mutations identified in the *ERG11* gene, as well as a higher number of copies of this gene and the overexpression of efflux pumps. For 5-fluorocytosine, it is hypothesized that the resistance is due to mutations in the *FCY2*, *FCY1,* and *FUR1* genes. Resistance to caspofungin has been associated with a mutation in the *FKS1* gene. Finally, resistance to each antifungal is closely related to the type of clade to which the strain belongs.

## 1. Introduction

*Candida auris* is characterized for being a critical pathogen worldwide that has resistance to virtually all antifungals commonly used in the treatment of invasive fungal infections. *C. auris* poses a severe threat to human health due to the increase in the frequency with which it occurs, mainly in Intensive Care Unit patients, where life-threatening candidiasis outbreaks have occurred. In some countries, this is being considered as a serious public health problem [1,2,3].

Since it was discovered in 2009, *C. auris* has spread very quickly. The first isolates came from Japan, India, Pakistan, South Africa, and Venezuela, but currently, its presence has been recorded in 19 other countries of the six continents [2,4,5].

Phylogenetic studies carried out in genomes of various *C. auris* [6,7,8] isolates have identified four predominant populations (clades I, II, III, and IV). The evidence of a smaller population, related to an isolate cultivated from the otic exudate of an Iranian girl, has probably identified a clade V [9]. While clades I to IV have persistently shown a wide geographical distribution, not many isolations have been obtained from the population of clade V [5].

The territorial expansion of the yeast, among other explanations, is because this organism has been identified incorrectly as *C. haemulonii, C. famata, C. catenulata,* or even *Rhodotorula glutinis* through phenotypic characterization techniques. This situation has favored the delay of preventive measures for contagion and dissemination or prevents them from being taken [3].

*C. auris* multi-resistance can be attributed to its phylogenetic relationship with other *Candida* species that are inherently resistant to more than one antifungal [3,5].

There are mainly three classes of antifungals for systemic use: azoles, polyenes, and echinocandins. It has been published that more than 90% of the *C. auris* isolates are resistant to fluconazole (azole), although resistance levels vary considerably between clades. Minimum inhibitory concentrations (MIC) that are high to amphotericin B (polyene) have been reported in several studies, and resistance to echinocandins is emerging in the isolates of some countries [10].

High rates of antifungal resistance in clade I have been observed, including four isolates resistant to the three main antifungal classes [5].

It has been reported that *Y132F* mutations in *ERG11* associated with azole resistance and *S639P* in *FKS1* for echinocandin resistance vary between clades. The copy number variations in *ERG11* predominate in clade III and have been associated with resistance to fluconazole [5].

It is essential to know the ways by which this organism presents its multi-resistance to improve health and control measures caused by *C. auris* infection.

The increase in the prevalence of multidrug-resistant *C. auris* to different groups of antifungals is a cause for alarm, especially considering that there are few treatment options nowadays. The latter has prompted the search for other therapeutic alternatives such as natural peptides, phenolic compounds, nitric oxide nanoparticles, as well as miltefosine and iodoquinol [11,12,13,14,15,16,17]. The conducted researches are promising, as they show that several of these alternatives can be used alone or in combination with other traditional antifungals, which produces a synergistic effect that potentiates fungal destruction.

## 2. Materials and Methods

The objective of this review is to provide recent information on the fundamental studies that explain from a molecular standpoint, the multi-resistance to antifungals that identifies most *C. auris* strains. To achieve this goal, the method used was a search that included the Science Direct website and the Springer Link, Wiley Online Library, and Medline (Pubmed) databases was performed, using as keywords: *Candida auris* antifungal resistance. The search yielded 4518 publications, of which 4415 were discarded for not meeting the subject or being in languages difficult to translate. Finally, only 102 articles were used, which were included in the 2000–2020 period, however, one article from 1994 was included due to its historical relevance. The review was performed based on PRISMA (Preferred Reporting Items for Systematic Reviews and Meta-Analysis) (Figure 1).

## 3. Polyenes

The drugs belonging to this group of antifungals are amphotericin B (AMB), nystatin, and natamycin. The latter acts in a different way from the AMB and the nystatin, as it interacts with ergosterol without forming ion channels; however, it can alter ergosterol-dependent reactions that precede the fusion of cell membranes [18].

Antifungal resistance to polyene is often rare in other *Candida* spp. species. (except *C. lusitaniae*) [19,20,21], but it can be generated through prophylactic or therapeutic exposure. Besides, it has been theorized that prior treatment with azoles decreases ergosterol concentrations resulting in subsequent resistance to polyenes [19,22,23]. Some studies show the complete sequencing of the *C. auris* genome, where three different amino acid substitutions have been observed in the *ERG11* gene related to different geographic clades (*F126T*, *Y132F*, *K143F*). Such mutations are associated with isolations in different continents; therefore, it can be inferred that the antifungal resistance that *C. auris* possesses can be acquired rather than being intrinsic [2,21,24,25].

Regarding the antifungal susceptibility of *C. auris* to polyenes, especially AMB, there are variable results according to different studies conducted in recent years; however, in most of them, it occupies the second place of resistance after fluconazole. Calvo et al., 2016 showed the in vitro susceptibility of 18 *C. auris* isolates with MIC50 of 1 mg/L and MIC90 of 2 mg/L, resulting in low susceptibility to AMB by having 50% of strains with elevated MICs [4].

It is suggested that between 10 and 35% of *C. auris* isolates are AMB resistant; for example, in the US, a study conducted by Ostrowsky et al., 2020 reported that nearly two-thirds of the studied isolates were resistant. On the other hand, in a study made by Vallabhareny et al., 2017, only one isolated of 7 was resistant [26,27,28]. In the systematic review of Osei Sekyere, 2018, 15.46% of the *C. auris* sample showed AMB resistance [29]. Arendrup et al., 2017, for their part, obtained that using the cut-off points given by the CDC, out of 742 *C. auris* isolates 111 were AMB resistant, corresponding to 15.46% which coincides with Osei Sekyere’s study [30]. In Kuwait, Khan et al., 2018 found that from a sample of 56 isolates, 13 were resistant to AMB and 11 to fluconazole, voriconazole, and AMB. However, there are no clinical cut-off points available for *C. auris.* Hence, if another MIC reference were considered for AMB, the resistance in this study would have increased significantly to 29 (52%) [31]. On the other hand, Lockhart et al., 2017 observed that out of 54 isolates, 19 were resistant to AMB, which represents 35% [6]. *C. auris* is therefore highly resistant to azoles and polyenes, which can be attributed to its phylogenetic relationship with other of *Candida* spp. species that are inherently resistant to multiple antifungals [6,29].

*C. haemulonii*, *C. duobushaemulonii,* and *C. pseudohaemulonii* may exhibit high-grade intrinsic resistance to AMB with MIC that can reach up to 16 mg/L [32]. Cases of acquired resistance to this group of antifungals have been reported in strains of *C. albicans*, *C. glabrata*, *C. rugosa*, *C. lusitaniae,* and *C. tropicalis;* as well as some cases of high MIC, poor response to treatment or even both in some strains of *C. albicans*, *C. krusei, C. rugosa*, *C. lusitaniae,* and *C. glabrata* as well as pan-resistant *C. auris* strains [22].

The molecular mechanism by which *C. auris* presents resistance to AMB is still unclear [1,33]. However, due to the mechanism of action of polyenes, alterations in the ergosterol pathway are considered the main resistance mechanism [10], which is related to *ERG2*, *ERG3*, and *ERG6* gene mutation [1,30,34,35,36,37]. It is possible that mutations can cause depletion and alteration of the ergosterol composition [36]. In in vitro studies with *C. albicans* and *C. glabrata* strains, mutations that disrupt these genes have been observed. In studies made with isolated AMB-resistant *C. auris* strains compared to susceptible strains, overexpression of the *ERG1*, *ERG2*, *ERG3*, *EGR5*, *ERG6*, and *ERG13* genes was observed [32,38]. Mutations in these genes lead to reduced ergosterol levels in the plasma membrane, and thus, with the decreased levels, the AMB resistance increases [20,23,28]. At the same time, there are also other studies such as the one from Muñoz et al., 2018 that contribute to the idea that antifungal resistance is rather given at the transcriptional level [39,40].

The study conducted by Muñoz et al., 2018 demonstrated that in *C. auris,* there is an intrinsic transcription of multidrug transporters higher than in other strains isolated from the study. After administrating AMB to one of the *C. auris* strains, five genes that are involved in the ergosterol biosynthesis path (*MVD*, *ERG2*, *ERG1*, *ERG6*, and *ERG13*) were induced. These genes are associated with the maintenance of the fungus cell membrane stability, similar to the transcriptional response occurring in *C. albicans*. Two strains of a clade had different resistance phenotypes than AMB, and, by further analyzing them, the induction of 8 genes that were positively regulated was achieved in both strains, including *OPT1*, *CSA1*, *MET15*, and *ARG1* [39]. Before polyene administration, one of the strains showed greater expression of genes involved in the transcriptional response of *C. albicans* to AMB, even when absent [39] (Table A1 in Appendix A).

A higher transcription of *CDR4a* was observed, suggesting that a greater intrinsic expression of multidrug transporters is a possible antifungal resistance mechanism in *C. auris* [39].

In an effort to understand how the molecular resistance mechanism to AMB is generated, Escandón et al., 2018 identified a single nucleotide polymorphism (SNPs) in a transcription factor, similar to that of *C. albicans,* called *FLO8* in an anonymous protein that supposedly encodes for a membrane transporter in AMB-resistant *C. auris* strains [32,41]. Escandón et al., 2019 were able to identify five non-synonymous mutations in protein-coding regions associated with AMB resistance in Colombian *C. auris* strains. Four mutations were found within the regions that encode for proteins, and one of them was the mutation in *FLO8*, a transcription factor required for virulence and formation of biofilms in *C. albicans,* which showed a change from a serine (in susceptible strains) to aspargine (in resistant strains). However, apparently, this mutation only represented a single event. No subsequent mutations occurred, and it has not been reported in the strains of *C. auris* 6684, B11220, B8441 from India, Japan, and Pakistan, respectively, and which are AMB-resistant [7,40,42]. Another mutation was found in a transmembrane protein that changes isoleucine to threonine, which could also contribute to antifungal resistance [7] (Table A1).

Thanks to the study of the *C. auris* genome, orthologous of virulence factors involved in the formation of biofilms, antifungal resistance, and phenotypic change have been identified, which are already known in *C. albicans* [1,20,24,29]. The formation of biofilms is especially suggested to have an important role in antifungal resistance in *C. auris*, including resistance to AMB, due to the possibility of being housed in this treatment-resistant cells which reduce drugs bioavailability [1,29,36,43,44]. Muñoz et al., 2018 showed in their study that *SIT1*, *PGA7*, and *RBT5* are positively regulated during the formation of biofilms in *C. auris*, thus suggesting that cell wall reorganization could be a response to antifungal treatment [39]. Sherry et al., 2017 showed that biofilms were resistant to doses > 4 mg/L of AMB and that only liposomal AMB was effective in limiting the growth at a lower concentration (0.25–1 mg/L); however, 16 mg/L were needed to stop biofilm activity at 90% [45].

Thanks to different studies, ABC and MFS type efflux pumps are considered to be involved in the antifungal resistance of *C. auris* [7,24,43,46,47]. Studies conducted by Ben-Ami et al., 2017 demonstrated through the use of rhodamine that *C. auris* had greater activity in ABC-type efflux pumps than other species such as *C. glabrata* and *C. haemulonii*, which can be associated with its participation as an antifungal resistance mechanism [46]. The latter coincides with research conducted by Chatterjee et al., 2015. They identified orthologous genes from ABC and MFS-type efflux pumps, which they consider to be a possible antifungal resistance mechanism to azoles, polyenes, and echinocandins [24] (Table A1).

Bhattacharya et al., 2020 propose replicative aging as another antifungal resistance mechanism in *C. auris*, as it origins stem cells and daughter cells that are phenotypically different, where old cells become more tolerant to AMB [34,44].

In the search for new treatment therapies for infections generated by *C. auris,* combined therapy of antifungals with essential oils has been proposed as phenolic compounds have been observed to present synergistic antifungal activity with antifungals, decreasing their MIC [17].

## 4. Triazoles

Triazoles are part of antifungal treatment against infections caused by various species of *Candida* due to their effectiveness against the formation of the fungal plasma membrane. However, their extensive use causes side effects in the host and resistance in the pathogen [48,49,50,51,52,53].

According to Lockhart, 2019, the main resistance mechanisms to triazoles exhibited by *C. auris* strains are (1) Mutations in the antifungal target, mainly in the *ERG11* gene, (2) Overregulation of *ERG11*, (3) Overregulation of efflux pumps. Likewise, this resistance is mostly towards fluconazole and less frequent towards other antifungals such as voriconazole, being that, from the evolution standpoint, it is a characteristic that arises on four occasions at least, so it is suggested that this process occurs due to the pressure exerted by the constant use of antifungals as a prophylactic method and for the treatment of nosocomial infections. This situation seems to be reinforced by the fact that fluconazole is the antifungal that is always available, which induces rapid mutations [6,32,54] (Table A1).

One of the most studied genes to understand the resistance of *C. auris* to antifungals is *ERG11*, which encodes for lanosterol-14α-demethylase (target protein of the fungistatic action of the triazoles) [26].

To the present date, 12 mutations have been identified in the *ERG11* gene, which are possibly determined by geographical area [6] and which decrease the affinity of a triazole with the target while reducing the susceptibility of the fungus to the drug. The latter also generates different susceptibility patterns to the triazoles depending on the strain’s place of origin [19,32,39,54,55].

In addition to mutations, overregulation of the gene may occur in azole-resistant strains, as some genomics and transcriptomic studies have observed an increase in *EGR11* transcripts, which may also be associated with cell age. However, this aspect has not been thoroughly studied yet [32,56].

Fungi, like other microorganisms, present groups of transport proteins called efflux pumps, which allow them to prevent the accumulation of high antifungals concentrations, thus decreasing their effectiveness. The most studied efflux pumps in *C. auris* are those that function based on transport proteins from the ABC (ATP Binding Cassette) and MFS (Major Facilitator Superfamily) families, which are determined by the expression of the *CDR1* gene [19,32,57].

In multi-resistant *C. auris* strains, it has been observed that ABC transport proteins are represented in a significant portion of the genome. Also, it has been seen that in the presence of fluconazole, a greater expression of *CDR1* occurs. Among the performed analyses, resistant strains that had the gene deleted were studied. The Minimum Inhibitory Concentration (MIC) of fluconazole and itraconazole decreased, so it is suggested that this is one of the mechanisms for triazole resistance [24,32,55,57,58] (Table A1).

Other resistance mechanisms that have been explored in recent years are related to the *HSP90* protein and chromosome duplication. However, both require further studies for full understanding.

The *HSP90* thermal shock protein is involved in the morphological changes of the fungus, suppressing the passage from the levaduriform phase to the filamentous phase of several species of *Candida*, especially *C. albicans* and *C. auris.* Kim et al., 2019, note that, in addition, this protein promotes resistance to triazoles by enabling key cellular responses that stabilize the cellular response to antifungal induced stress, since inhibition of the expression of this protein reduces the tolerance of the strain to fluconazole, by allowing its action to transform from fungistatic to fungicide, decreasing the survival of the fungus. However, few studies have been conducted on this subject [59] (Table A1).

Another mechanism of resistance to the triazoles is the duplication of the chromosome on which the *ERG11* gene is located. However, the correct functioning of this process must still be determined [32,56]. This mechanism was observed only in stem cells that subsequently divide, and then it disappears into the daughter cells [56].

## 5. 5-Fluorocytosine

The treatment of infections caused by different *Candida* species with 5-FC has shown effectiveness, except in *C. glabrata* and *C. krusei* [60]; however, in *C. auris* data is scarce and variable. Schelenz et al., 2016 reported that 15 clinical and environmental isolates from a *C. auris* outbreak that occurred at a cardio-thoracic centre in London, were susceptible to 5-FC (MIC < 0.06–0.12 mg/L) [61]. In the same way, Lockhart et al., 2017 assessed susceptibility to antifungals in 54 *C. auris* isolates and found susceptibility with MIC values in the range of 0.125–0.5 mg/L [62,63]. On the other hand, Chowdhary et al., 2014 reported 47% resistance to 5-FC (MIC ≥ 64 mg/L) in 15 *C. auris* clinical isolates, while Kathuria et al., 2015 found that out of 90 *C. auris* isolates that were analyzed, 12% showed resistance to 5-FC with MIC (≥32 mg/L) and 88% showed MIC susceptibility (0.4 mg/L) [60,62]. Ben-Ami et al., 2017 analyzed six *C. auris* isolates, of which five reported susceptibility to 5-FC (0.25–0.5 mg/L) and one not susceptible (1 mg/L) [46]. Osei Sekyere, 2018 calculated the overall resistance rate to 5-FC from reports of 742 clinical isolates of *C. auris* in 16 countries, mainly India, the United States and the United Kingdom, which resulted in 1.95%, with MIC values between 1–128 mg/L [29]. Recently, O’Brien et al., 2020 analyzed 277 *C. auris* clinical isolates, finding a low ratio (0.7%) isolates resistant to 5-FC (MIC ≥ 32 mg/L) [64]. Based on these reports, it can be said that 5-FC is very active against *C. auris*, but it is clear that the risk of the yeast developing resistance persists.

It has been reported that fungal cells may develop resistance to 5-FC as a result of mutations that occur mainly in genes involved in the absorption and metabolism of 5-FC: *FCY2*, which encodes the purine-cytosine permease; *FCY1*, which encodes the cytosine deaminase, and *FUR1* encoding the uracil phosphoribosyl transferase [44,65]. Resistance can be primary when related to a decrease in drug absorption by mutations in the gene *FCY2*, and secondary when there is a limitation in the conversion of 5FC into 5FU, or 5-fluorouridine monophosphate (5-FUMP) by alterations in *FCY1* and *FUR1* genes respectively [66]. In *C. auris*, there is no information regarding 5-FC resistance mechanisms; there is only one report of an isolate with the F211I mutation in an isolate that reported high MIC values in vitro, this mutation causes the substitution of isoleucine to phenylalanine [8] (Table A1). Although no further studies have been conducted to correlate the mutation with resistance to 5-FC, this is very likely a pathway through which *C. auris* acquired resistance to the antifungal, since mutations have been identified as an important resistance mechanism to 5FC in *C. albicans,* and *C. lusitaniae* with whom it has a close phylogenetic relationship [67,68,69]. Dogson et al., 2004 determined that a single change of nucleotide, from cytosine to thymine at position 301 of the *FUR1* gene, which results in a change from arginine to cysteine at position 101 of the amino acid chain, is the most important mechanism of resistance to 5-FC, since yeasts with this mutation had VALUES of ICM 16 mg/mL to 5-FC [68].

Papon et al., 2007 analyzed the inactivation of genes *FCY2*, *FCY1* and *FUR1* in *C. lusitaniae* and found that mutations in *FUR1* generate resistance to 5-FU with high MIC values of (≥512 mg/L), while mutations in *FCY1* or *FCY2* confer cross-resistance to 5-FC and FLC, with a lower resistance level to 5-FC (MIC = 64–128 mg/L) compared to *FUR1* inactivation [69].

To minimize the risk of therapeutic failure caused by the ability with which fungi develop resistance to 5-FC, it is recommended to use this drug in combinations with other classes of antifungals instead of monotherapy, as this has shown to promote a synergistic interaction between molecules [70]. Zhu et al., 2020 reported that combinations of 5-FC with other antifungals are successful in treating *C. auris* infections: for nine *C. auris* isolates resistant to AMB (>2.0), the combination of AMB/5-FC (0.25/1.0124 mg/L) produced 100% inhibition. Six *C. auris* isolates resistant to three echinocandins ([AFG > 4.0 mg/L], [CAS > 2.0 mg/L], [MFG > 4.0 mg/L]), were 100% inhibited by the AFG/5-FC and CAS/5-FC (0.0078/1 mg/L) combinations, and MFG/5-FC (0.12/1 mg/L). Thirteen isolates with high MIC values for VRC (MIC > 2 mg/L) were 100% inhibited when the combination VRC/5FC (0.015/1 mg/L) was used [71].

## 6. Echinocandins

Echinocandins are synthetically modified lipopeptides that uncompetitive inhibit the β-1,3-d-glucan synthase, which is the enzyme responsible for the biosynthesis of the main structural element of the fungal cell wall [71,72,73,74]. There are several types of echinocandins; the caspofungin was the first U.S. FDA-approved echinocandin, and it is commercially available as caspofungin acetate. The next echinocandins added to the list were micafungin and anidulafungin, which were approved in 2005 and 2006, respectively [75,76].

The determination of resistance to echinocandins can be performed phenotypically using the broth microdilution or disc diffusion methods, the Etest method, as well as commercial systems. It can also be done molecularly by detecting mutations in the “hot spots” (HS) regions of *FKS1* and *FKS2* genes. In addition, the Matrix-Assisted Laser Desorption Ionization Time-of-Flight Mass Spectrometry (MALDI-TOF) technique for echinocandin resistance detection is already in use [77,78,79,80,81,82]. Two organizations, the European Committee on Antimicrobial Susceptibility Testing (EUCAST) and the Clinical and Laboratory Standards Institute (CLSI), have established reproducible and highly reliable susceptibility broth dilution tests for *Candida* spp., and echinocandins. These reference methods have shown to be reliable and useful in clinical mycology reference laboratories for determining resistance in clinical isolations [83,84,85].

Resistance to echinocandins can lead to clinical failures and is conferred by nucleotide substitutions that generate amino acid changes in the catalytic subunit of the target enzyme of echinocandins, the β-1,3-d-glucan synthase, which is encoded by *FKS1*, *FKS2* and *FKS3* genes [86]. The mechanism is highly specific and different from the azole group, because resistance to echinocandins is not affected by multidrug transporters [87]. Mutations in *FKS* that confer resistance to echinocandins are located in 2 highly conserved regions of the *FKS1* gene known as “hot spot” (HS) that include Phe641-Pro649 and Arg1361 residues and in homologous regions of the *FKS2* gene in *C*. *glabrata* (Table A1). 

For *C*. *albicans*, amino acid changes in Ser641 and Ser645 are the most common ones and cause a higher resistance phenotype, while in *C*. *glabrata*, amino acid modifications in Ser663 in *FKS2*, Ser629 in *FKS1* and Phe659 in *FKS2* are the most common amino acid substitutions [88,89,90,91]. Finally, Johnson et al., 2011 reported a third hot-spot region in W695L in *FKS1* generated through site-directed mutagenesis in *Saccharomyces cerevisiae* from an in vitro study, but this mutation has not been observed in clinical isolations [92]. Amino acid substitutions may decrease the sensitivity of glucan synthase, resulting in high minimum inhibitory concentration (MIC) values. 

Information on the molecular resistance mechanisms of *C*. *auris* to different antifungal agents is scarce, and the accuracy of these mechanisms in the different isolated strains is not well defined. *C. auris* escapes from the microbicidal effect of most antifungal groups through different mechanisms, including mutations in *ERG3* and *ERG11* genes, positive gene regulation of the efflux pump and single nucleotide polymorphism (SNP) in different genome loci [6,27].

As in other *Candida* species, resistance to echinocandin in *C*. *auris* is associated with mutations in HS sites of the *FKS1* gene, which encodes the catalytic subunits of the target enzyme β-1,3-d-glucan synthase [86] (Table A1). Resistance to echinocandins has been reported mainly in *C*. *auris* isolates from India and South Africa in studies conducted by Chowdhary et al. 2014, 2018 and Sharma, 2016, in which 2% of the isolates had a high MIC of ≥8 mg/L (pan-echinocandin resistant phenotype) against caspofungin, micafungin and anidulafungin. On the other hand, Hou et al., 2018 analyzed the HSI of *FKS1* in 4 isolations that presented serine in place of phenylalanine in codon 639 (S639F), which is equivalent to the S645 substitution in *FKS1* HSI in *C*. *albicans* [19,47,62,93]. Also, Berkow et al., 2018 assessed in a drug response test in vivo and did not respond to echinocandin therapy in the invasive candidiasis mouse model [94]. A different amino acid substitution in the same position (S639P) of the HSI was recently reported by Berkow et al., 2018 in *C*. *auris* isolates resistant to echinocandin, which corresponds to the S645P substitution in *C*. *albicans*, and S629P *C*. *glabrata* [94]. The last mutation in HSI associated with equinocandin resistance is S639Y [62].

Due to the increasing prevalence of *C*. *auris* multidrug resistance, surveillance is recommended in patients infected or colonized with *C*. *auris* [55]. The increase in multi-resistant strains to different antifungal groups is a global alarm as it leaves few therapeutic options and becomes a global public health problem.

Recently, the rezafungin (formerly CD101), which is in final evaluation tests (phase 3), has been integrated into this group. This new member is a more stable structural analog of the anidulafungin as it possesses half a choline at the C-5 position of ornithine. It has a prolonged half-life of approximately 130 h, which is very useful from the clinical standpoint for weekly intravenous administrations. However, the EUCAST and CLSI cut-off values are not yet defined [95,96].

There are some reports of the rezafungin behavior against *C*. *auris* isolates from different countries such as Pakistan, Venezuela, Panama, Colombia, India, South Africa, Israel, and the United States. In vitro activity was very similar for all isolates with average MIC values of 0.25 mg/L. Regarding isolates that had mutations in *FKS1* HS1, the observed rezafungin MIC values were slightly lower than the MIC values of anidulafungin and micafungin in most *Candida* species; however, they were noticeably smaller in *C*. *auris* isolates [6,94,97,98].

## 7. Alternative Treatment Strategies in *C. auris*

The high rate of antifungal resistance of *C. auris*, and its great ability to form biofilms, creates the need to look for alternative treatments for the infections caused by this yeast. Among the alternatives that have been explored are natural peptides, phenolic compounds, nitric oxide in nanoparticles, as well as the reuse of drugs such as miltefosine and iodoquinol [11,12,13,14,15,16,17,99]. One of the natural peptides with antimicrobial activity is crotamine, a toxin obtained from the rattlesnake’s venom [11]. The antifungal effect of crotamine was evaluated by Dal Mas et al., 2019 in *Candida* spp. clinical isolates, including fluconazole and amphotericin-B-resistant *C. auris*. Crotamine showed a fungicide effect in vitro at a concentration of 40–80 µM. Based on these results, it has been proposed that the crotamine’s chemical structure can serve as a model for the generation of new antifungals against *C. auris* and other multi-resistant *Candida* species.

Plant essential oils are of great interest, as they contain various phenolic compounds with fungicide activity against *Candida* species. Among these compounds are eugenol, methyl eugenol, thymol, carvacrol, and farnesol, with carvacrol and farnesol being the most active compounds [15,17]. Shaban et al., 2020 conducted in vitro studies, where carvacrol showed a MIC value of 125 mg/L, and its combination with fluconazole, amphotericin B, nystatin and caspofungin produced synergistic and additive effects at 68%, 64%, 96%, and 28%, respectively. Besides, the combination of carvacrol reduced the MIC values of antifungals [17]. Farnesol has shown significant inhibition of planktonic cell growth in *C. auris* at concentrations ranging from 0.3 to 62.5 mM within 24 h [15,100]. With regards to the in vivo experiments, daily treatment with 75 mM of farnesol decreased the fungal load in an immunosuppressed murine model with disseminated candidiasis, especially in the case of pre-exposed inoculums to farnesol [15]. Both carvacrol and farnesol inhibit cell adherence and biofilm formation, mitigating *Candida* spp. virulence [15,17,100]. Based on this evidence, phenolic compounds can be said to have an alternative or adjuvant therapeutic potential in conventional treatments to control life-threatening infections caused by *C. auris*.

Cuminaldehyde is another essential oil component that has also been used to develop eleven azolic compounds (UoST1-11) by replacing the toxic aldehyde group with isopropyl or ter-butyl at the C-4 position of the phenyl ring and the replaced phenyl bonded to the *N*-4position of the triazole ring bearing 3-Cl, 4-Cl, 4-CH3, 4-OCH3, 4-F, 3-Br or a non-replaced phenyl [13]. Of the eleven compounds, UoST5, 7, 8, and 11 had significant antifungal activity against *C. auris* and other multidrug-resistant pathogens. UoST5 was also formulated in polymeric nanoparticles (NPs) to mitigate their toxicity. With the new formula, UoST5-NPs provided a 25% release after 24 h, thus maintaining a prolonged activity up to 48 h, without presenting toxicity in low concentrations [13].

On the other hand, nitric oxide (NO), a natural product of the immune system, acts as a broad-spectrum antimicrobial agent. In fungi, the NO has shown efficacy against *C. auris* [12,101]. Cleare et al., 2020 demonstrated the effectiveness of NO nanoparticles (NAC-SNO-np) against six *C. auris* clinical isolates. This nanoparticle formulation slowly releases NO, *N*-acetylcysteine (NAC), and N-acetylcysteine S-nitrosothiol (NAC-SNO), which exhibit antimicrobial activity. The NO released is supplied continuously to the *C. auris* planktonic cells to inhibit their proliferation; it also significantly disrupts the biofilms formation. Based on the later, NAC-SNO-np represents a promising alternative to prophylactic and therapeutic treatments for *C. auris* infections [12].

A quick and cost-effective alternative to counteract *C. auris* is the reuse of existing medicines for the treatment of other diseases as antifungals. The most representative examples are iodoquinol and miltefosine (MFS) [14,99]. Iodoquinol, a halogenated derivative of quinoline, is used as an intestinal antiparasitic drug, as well as to treat bacterial and fungal skin infections [99]. Iodoquinol has significantly shown in vitro inhibitory activity against *C. auris* planktonic cells; however, its effect on biofilms has been negligible [99]. The MFS, an antineoplastic and antiparasitic drug that, in free form or encapsulated in alginate nanoparticles (MFS-AN) to reduce its gastrointestinal toxicity, has shown fungicide activity (2 to 4 mg/L) in *C. auris* clinical isolates. Also, it inhibits the formation of biofilms at a concentration of 0.25 to 4 mg/L. In comparison, at a higher concentration (16 to 32 mg/L), it prevents the growth of preformed biofilms [14]. During in vivo trials, treatment with free MFS or MFS-AN has significantly improved the survival and morbidity rates of *Galleria mellonella* larvae infected by *C. auris*, which demonstrates that MFS is a promising therapeutic alternative [14].

Finally, with the evidence that nanoencapsulation strategies reduce drug toxicity, Rodriguez et al., 2020 developed micellar systems for the administration of amphotericin B, using amphiphilic block copolymers (ABCs) conjugated with retinol. Amphotericin B encapsulated in polymeric micelles has shown an improvement in antifungal efficiency, with MIC values of 0.93–1.865 mg/L against *C. albicans* and *C. auris*, compared to the MIC values (3.75 mg/L) of Fungizone^®^. Thus, it is possible that infections caused by amphotericin-B-resistant *C. auris* are treated with the same antifungal but encapsulated in polymeric micelles [16].

## 8. Conclusions

*Candida auris* is considered a global public health problem due to the increase of multi-resistant strains to different antifungal groups, leaving few therapeutic options currently. The possibility has been raised that the *C. auris* genome has adapted to emerge as a resistant pathogen due to the indiscriminate use of broad-spectrum antibiotics and antifungals, as it presently occurs with resistance to triazoles, due to their abuse in marginalized populations. At the same time, the study of molecular resistance mechanisms to different kinds of antifungals has become complicated due to the genetic variation between *C. auris* clades. As it is known, resistance to polyenes varies depending on the region where the strain is isolated; likewise, reports of *C. auris* strains resistant to echinocandins have been made in India and South Africa. For now, the knowledge of the molecular resistance mechanisms of *C. auris* is scarce; however, this resistance can occur through mutations in the ERG2, ERG3, ERG6, ERG11, and FKS1 genes, positive regulation of the efflux pump genes and single nucleotide polymorphism (SNP) in different *loci* of its genome. The use of polyenes for the treatment of *C. auris* can be enhanced by using other antifungals such as 5-FC. Although 5-FC doesn’t constitute the treatment of choice for these infections, it is a good option if given in combination with polyenes, echinocandins, or azoles. *C. auris* is a widely spread organism that causes a high mortality rate; however, this species has been little studied. It is imperative to explore in-depth the resistance mechanisms for this emerging yeast, as well as research aimed at other molecular factors that may be involved in tolerance and resistance to antifungals to find new and better treatment options.

It is important to note that the high rate of antifungal resistance of *C. auris* has prompted the search for alternative treatment strategies; however, it is necessary to conduct thorough research before they can be applied in patients. We thank Shiftext for the translation.

## Figures and Tables

**Figure 1 antibiotics-09-00568-f001:**
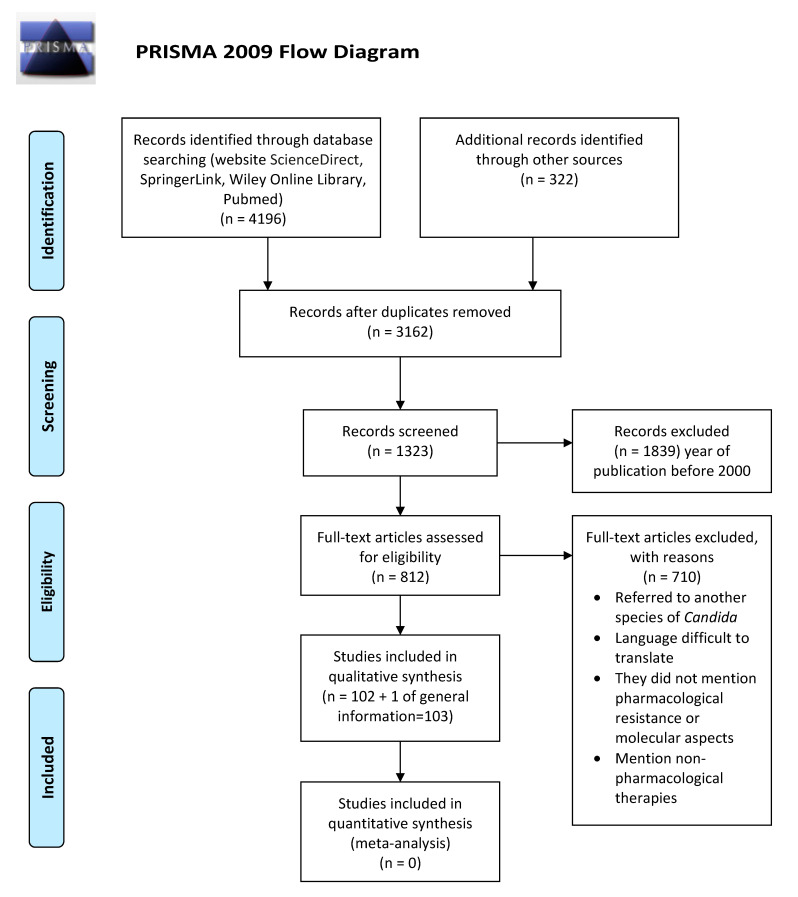
Flowchart of the different phases of the systematic review.

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
