# Peer review of "Antifungal Resistance in Candida auris: Molecular Determinants"

_antibiotics, 2020, doi:10.3390/antibiotics9090568_

Round 1
Reviewer 1 Report
In this reviewed manuscript, the authors have summarized the latest findings on molecular mechanisms of antifungal resistance in Candida auris. The manuscript is interesting and well written so I have one comment/suggestion to authors.
It will be a more comprehensive review, if authors will add a section about experimental alternative therapies other than polyene, triazoles, echinocandins and 5-fluorocytosine used or proposed for C. auris treatment such as phenolic compounds ( flavonoids etc) or nanoparticles or natural compounds isolated from toxins
Author Response
We want to thank the two anonymous reviewers for taking the time to read our paper and for providing valuable feedback. The insightful comments and suggestions will undoubtedly improve the manuscript. We have revised the content accordingly and have incorporated the changes as suggested. Please find our response to the reviewer's specific comments highlighted in gray. Modifications in the manuscript text are marked in yellow.
Reviewer 1
English language and style
( ) Extensive editing of English language and style required
( ) Moderate English changes required
(x) English language and style are fine/minor spell check required
( ) I don't feel qualified to judge about the English language and style
|
Is the work a significant contribution to the field? |
|
|
Is the work well organized and comprehensively described? |
|
|
Is the work scientifically sound and not misleading? |
|
|
Are there appropriate and adequate references to related and previous work? |
|
|
Is the English used correct and readable? |
Comments and Suggestions for Authors
In this reviewed manuscript, the authors have summarized the latest findings on molecular mechanisms of antifungal resistance in Candida auris. The manuscript is interesting and well written so I have one comment/suggestion to authors.
It will be a more comprehensive review, if authors will add a section about experimental alternative therapies other than polyene, triazoles, echinocandins and 5-fluorocytosine used or proposed for C. auris treatment such as phenolic compounds ( flavonoids etc) or nanoparticles or natural compounds isolated from toxins
Answer: We thank the reviewer for the kind comments. A section called "Alternative treatment strategies in C. auris" was added to the manuscript to address this suggestion.

Reviewer 2 Report
María Guadalupe Frías-De-León and her collaborators wrote a review about the current knowledge of antifungal resistance in C. auris. Although the described topic is important the quality of manuscript is not too high. The coherency of MS is very low so it should be improved. The Authors wrote about four antifungal classes; however, the style of description of these sections is very different and it is disturbing. In addition, there are more unnecessary information about general properties of various antifungals, which are well known for target audience.
- You should remove the first 17 lines from the part of polyenes
- You should remove the first 16 lines from the part of triazoles
- You should remove the first 6 lines form the part of 5FC
- You should remove the first 28 lines from the part of candins
This unnecessary information should be summarized in one-one sentence per section.
Furthermore, I suggest a separate section about the potential traditional and/or alternative treatment strategies against multi-resistant C. auris.
Minor points:
The writing style of species name, abbreviations etc. is not homogenous in this review. You must improve it. For example: C. Albicans vs. C. albicans; mg/L vs Mg/L
In the introduction, you should refer to the potential fifth clade derived from Iran.
The last sentence in introduction is not proper at that point, please remove it.
„Polyenes (amphotericin B, nystatin, natamycin)” You listed the polyenes in the section title but not triazole and candins. It is disturbing.
You should write about the rezafungin vs auris in the part of echinocandins (e.g.: PMID: 32015032, PMID: 31539426 etc.)
Author Response
Answers to Reviewers
We want to thank the two anonymous reviewers for taking the time to read our paper and for providing valuable feedback. The insightful comments and suggestions will undoubtedly improve the manuscript. We have revised the content accordingly and have incorporated the changes as suggested. Please find our response to the reviewer's specific comments highlighted in gray. Modifications in the manuscript text are marked in yellow.
Reviewer 2
English language and style
( ) Extensive editing of English language and style required
( ) Moderate English changes required
( ) English language and style are fine/minor spell check required
(x) I don't feel qualified to judge about the English language and style
|
Is the work a significant contribution to the field? |
|
|
Is the work well organized and comprehensively described? |
|
|
Is the work scientifically sound and not misleading? |
|
|
Are there appropriate and adequate references to related and previous work? |
|
|
Is the English used correct and readable? |
Comments and Suggestions for Authors
María Guadalupe Frías-De-León and her collaborators wrote a review about the current knowledge of antifungal resistance in C. auris. Although the described topic is important the quality of manuscript is not too high. The coherency of MS is very low so it should be improved. The Authors wrote about four antifungal classes; however, the style of description of these sections is very different and it is disturbing. In addition, there are more unnecessary information about general properties of various antifungals, which are well known for target audience.
- You should remove the first 17 lines from the part of polyenes
Answer: The lines were removed accordingly.
- You should remove the first 16 lines from the part of triazoles
Answer: The lines were removed accordingly.
- You should remove the first 6 lines form the part of 5FC
Answer: The lines were removed accordingly.
- You should remove the first 28 lines from the part of candins
Answer: The lines were removed accordingly.
This unnecessary information should be summarized in one-one sentence per section.
Answer: We thank the reviewer for pointing out this issue. The information was rewritten to improve clarity and conciseness.
Furthermore, I suggest a separate section about the potential traditional and/or alternative treatment strategies against multi-resistant C. auris.
Answer: A section called "Alternative treatment strategies in C. auris" was added to the manuscript to address this suggestion.
Minor points:
The writing style of species name, abbreviations etc. is not homogenous in this review. You must improve it. For example: C. Albicans vs. C. albicans; mg/L vs Mg/L
Answer: The writing style was corrected in the manuscript.
In the introduction, you should refer to the potential fifth clade derived from Iran.
Answer: This information has been added to the introduction.
The last sentence in introduction is not proper at that point, please remove it.
Answer: This sentence has been removed.
"Polyenes (amphotericin B, nystatin, natamycin)" You listed the polyenes in the section title but not triazole and candins. It is disturbing.
Answer: We appreciate this observation. The title has been changed to "Polyenes" for the sake of consistency.
You should write about the rezafungin vs auris in the part of echinocandins (e.g.: PMID: 32015032, PMID: 31539426 etc.)
Answer: This information has been added to the echinocandins section.

Round 2
Reviewer 2 Report
The manuscript has significantly improved. Nevertheless, I have some minor issues, which should be revised yet.
„Echinocandins are a new weapon integrated into the group”
New??? Caspofungin was introduced in 2001 and the last one was introduced in 2007. (Rezafungin is in Phase 3 now). Please reformulate this sentence.
„Farnesol has shown significant inhibition of planktonic cell growth in C. auris at concentrations ranging from 62.5 to 300 mM within 24 h [15,102]. With regards to the in vivo experiments, daily treatment with 75 mM of farnesol decreased the fungal load in an immunosuppressed murine model with disseminated candidiasis, especially in the case of pre-exposed inoculums to farnesol”
The dimension of farnesol is micromol and NOT milimol in paper published by Nagy et al. Please correct it.
Author Response
Answers to Reviewers
We want to thank once more the anonymous reviewer for the thorough revision. Please find our response to the reviewer's specific comments highlighted in gray. The manuscript was updated accordingly to reflect the requested changes.
Reviewer 2
English language and style
( ) Extensive editing of English language and style required
( ) Moderate English changes required
( ) English language and style are fine/minor spell check required
(x) I don't feel qualified to judge about the English language and style
|
Is the work a significant contribution to the field? |
|
|
Is the work well organized and comprehensively described? |
|
|
Is the work scientifically sound and not misleading? |
|
|
Are there appropriate and adequate references to related and previous work? |
|
|
Is the English used correct and readable? |
Comments and Suggestions for Authors
The manuscript has significantly improved. Nevertheless, I have some minor issues, which should be revised yet.
"Echinocandins are a new weapon integrated into the group"
New??? Caspofungin was introduced in 2001 and the last one was introduced in 2007. (Rezafungin is in Phase 3 now). Please reformulate this sentence.
Answer: We agree with this observation. The sentence was removed to avoid inaccuracies.
"Farnesol has shown significant inhibition of planktonic cell growth in C. auris at concentrations ranging from 62.5 to 300 mM within 24 h [15,102]. With regards to the in vivo experiments, daily treatment with 75 mM of farnesol decreased the fungal load in an immunosuppressed murine model with disseminated candidiasis, especially in the case of pre-exposed inoculums to farnesol"
The dimension of farnesol is micromol and NOT milimol in paper published by Nagy et al. Please correct it.
Answer: The dimensions were corrected to match the reference.
